# Analysis of Protein-Protein Interactions for Intermolecular Bond Prediction

**DOI:** 10.3390/molecules27196178

**Published:** 2022-09-21

**Authors:** Justin Z. Tam, Talulla Palumbo, Julie M. Miwa, Brian Y. Chen

**Affiliations:** 1Department of Computer Science and Engineering, Lehigh University, Bethlehem, PA 18015, USA; 2Department of Biological Sciences, Lehigh University, Bethlehem, PA 18015, USA

**Keywords:** intermolecular bond prediction, bond classifier, DiffBond, ionic bond identificatio

## Abstract

Protein-protein interactions often involve a complex system of intermolecular interactions between residues and atoms at the binding site. A comprehensive exploration of these interactions can help reveal key residues involved in protein-protein recognition that are not obvious using other protein analysis techniques. This paper presents and extends DiffBond, a novel method for identifying and classifying intermolecular bonds while applying standard definitions of bonds in chemical literature to explain protein interactions. DiffBond predicted intermolecular bonds from four protein complexes: Barnase-Barstar, Rap1a-raf, SMAD2-SMAD4, and a subset of complexes formed from three-finger toxins and nAChRs. Based on validation through manual literature search and through comparison of two protein complexes from the SKEMPI dataset, DiffBond was able to identify intermolecular ionic bonds and hydrogen bonds with high precision and recall, and identify salt bridges with high precision. DiffBond predictions on bond existence were also strongly correlated with observations of Gibbs free energy change and electrostatic complementarity in mutational experiments. DiffBond can be a powerful tool for predicting and characterizing influential residues in protein-protein interactions, and its predictions can support research in mutational experiments and drug design.

## 1. Introduction

Chemical bonds play a crucial role in the interaction of proteins. Understanding intermolecular bonds in particular is an important part of discovering how protein-protein complexes achieve selective binding. In structural biology, the process of deducing the role of chemical bonds requires a multi-step geometric and chemical analysis. First, bonds are identified from the structure of a protein-protein complex by applying appropriate geometric and chemical constraints. Next, hypotheses are developed about the role that bonds play in the protein complex—whether the bond stabilizes the complex, assists in the binding process, or plays a different role. Finally, these hypotheses are tested in a range of experiments, such as by mutation, which alter the specific residues involved in the bond and result in changes in binding affinity. Computational methods can support the first two parts of this process, with a focus on limiting the number of necessary experiments.

This paper aims to assess the predictability of intermolecular bonds while applying standard definitions of bonds in chemical literature to explain protein interactions. Our approach is to apply the textbook chemical measurements and constraints of ionic interactions, hydrogen bonds, and salt bridges at biological pH as a predictor for the presence of intermolecular bonds, and verify these predictors against experimental findings established from literature. Measurements of hydrogen bonds are extremely well defined [1], but standard chemical measurements of ionic bonds and salt bridges are less so. Attraction and repulsion can occur between any charged atoms at any distance based on Coulomb’s law, however the degree of attraction or repulsion depends significantly on the geometry and nature of the dielectric between the charged atoms. As a result, ionic bond measurements must always be an approximation based on assumptions of the biological environment, as examined by several groups [2,3]. Chemical literature defines salt bridges as the co-occurrence of both a hydrogen bond and an ionic bond; salt bridge measurements must also exist under the same biological assumptions as those of ionic bonds. Following these definitions, we extend upon [4] to present DiffBond, a method for identifying and classifying intermolecular bonds while applying standard definitions of bonds in chemical literature to explain protein interactions. This paper paraphrases significantly the implementation and verification of DiffBond and also adds additional verification of DiffBond with affinity data.

A broad range of techniques have been developed for identifying different characteristics of protein complexes that influence protein-protein interactions. Some methods analyze mutation stability from heuristic energy changes [5], rigidity-based structure analysis [6], or molecular dynamics simulations [7]. Other methods employ computational models to evaluate electrostatic potential volumes [8] or predict hydrogen bond locations [1] or salt bridge locations [2,3] from crystal structure. In this context, Diffbond applies these methods and concepts to contribute a unified tool for finding hydrogen bonds, ionic bonds, and salt bridges in a protein structure. This capability makes bond analysis more accessible to non-experts in biochemistry and more automated for larger scale analysis.

DiffBond is the first to classify intermolecular bond interactions across the binding interface between protein-protein interactions into salt bridge, ionic bond, or hydrogen bond categories. For hydrogen bonds, DiffBond uses HBPlus for hydrogen bond identification [1]. For salt bridges, we applied previous definitions developed by [2]. For ionic bonds, we developed a definition from standard chemical measurements since no explicit geometric definition has been developed for structural analysis, to our knowledge. Our implementation of DiffBond follows a bottom-up approach for discovering each type of intermolecular interaction; we identify each interaction independently using three distinct methods that can work separately and in combination of one another.

DiffBond contributes to a larger Analytic Ensemble strategy. An Analytic Ensemble strategy, mentioned first in [9], aims to explain the biochemical mechanisms that achieve specific binding by creating tools that exclusively examine one mechanism. In earlier work, we have demonstrated techniques that exclusively identify steric [10,11] and electrostatic [8,9] influences on specificity. Their exclusivity means that influences they identify must act through steric or electrostatic mechanisms, respectively, since they examine nothing else. This deductive approach means that Analytic Ensembles can generate true biochemical explanations that elucidate real specificity mechanisms, as long as the mechanisms themselves are examined by underlying tools. DiffBond enables the identification of bonds for a future analysis of this kind.

## 2. Materials and Methods

DiffBond predicts hydrogen bonds, ionic bonds, and salt bridges using separate methods. In this section, we describe how each bond is identified from these coordinates, beginning with the atomic coordinates of a protein structure. Following that, we explain experimental methods used to verify these techniques.

### 2.1. Intermolecular Bond Searching

#### 2.1.1. Hydrogen Bonds

Hydrogen bonds are found using HBPlus [1], which measures several strict criteria of hydrogen bonds to predict hydrogen bond formation. First we identify a donor, an atom covalently bound to a hydrogen, and an acceptor, an atom with a lone pair of electrons that forms the hydrogen bond with the donor hydrogen. The specific identification of donors and acceptors is referenced from chemical literature for each amino acid. The biochemical distance constraints require that a donor must be within a 3.9 Å distance of an acceptor and that a donor hydrogen must be within 2.5 Å from the acceptor. The angle formed from the donor-hydrogen-acceptor interaction must also be greater than 90°, and the donor and acceptor must be a minimum of three covalent bonds apart. These measures ensure that the donor atom is further from the acceptor atom than the donor hydrogen and that the bond is not affected by steric hindrance. The angle formed between the hydrogen-acceptor and acceptor antecedents, neighboring atoms of the acceptor atom, as well as the angle between donor-acceptor-acceptor antecedent must be a minimum of 90°, to maintain biologically relevant hydrogen bond conformations between interacting side chains. These parameters have been used in previous hydrogen bond interaction studies with evidence that these values yield high specificity [1,12,13].

#### 2.1.2. Ionic Bonds

Intermolecular ionic bonds are found through a search algorithm for residues within a distance and electrostatic criteria. A distance less than 5 Å between two amino acid centroids is used as a standard measurement for possible interactions. A cutoff distance of 5 Å is strict enough to yield most biochemically relevant residues. Ionic bonds, rarely, can form over longer distances between 5–10 Å [14], and so we also provide ionic bond predictions at 7.5 Å and 10 Å in Appendix A. Electrostatically, residues must be oppositely charged amino acids at physiological pH to form an ionic bond, namely interactions between positively charged arginine, histidine or lysine with aspartate or glutamate [15]. An additional constraint requires the charged atoms of the amino acid, either a positive N (nitrogen) in basic residues or a negative O (oxygen) in acidic residues, are within the distance cutoff. By applying such a constraint, it forces residue side chains to be oriented towards each other in the model and be at a distance where interactions by proximity can be assumed despite the side chain orientation.

#### 2.1.3. Salt Bridges

Intermolecular salt bridges are identified using a similar procedure to that of ionic bonds. Salt bridge distance and electrostatic measurements are measured between oppositely charged residues whose side chain charged atoms, N and O, are interacting within a distance of 4 Å [3]. Ref. [16] also established 4 Å distance as a well defined distance cutoff that yields biochemically relevant salt bridge geometry. A side effect of using a 4 Å distance cutoff is that oppositely charged atoms within this distance are unable to accommodate a water molecule in between them, which further supports the possibility of hydrogen bond formation and simplifies the environmental conditions surrounding salt bridge prediction [2,17]. As a result, the salt bridge constraints reflect a stricter variant of the ionic bond and demonstrates electrostatic relevance of ionic interactions and geometric relevance of hydrogen bonds. Figure 1 shows an example of oppositely charged side chains, glutamate and arginine of a Barnase-Barstar protein complex, within 4 Å distance of each other.

This procedure of searching for residue pairs bound by the respective constraints of each type of intermolecular bond, allows us to identify relevant bond predictions. The implementation of DiffBond does not require dimeric protein complexes; higher order oligomers consisting of many subunits or protein-protein interactions involving multiple proteins can be assessed in a similar way as dimers without losing interface information or violating assumptions.

### 2.2. Electrostatic Isopotential Surfaces

An alternate method from bond prediction for finding influential amino acids is by interpreting protein electrostatic fields using VASP-E [8]. An electrostatic field of a protein represents the accumulation of charge from charged residues throughout a protein. By interpreting the electrostatic potential field between protein-protein interactions, especially of oppositely charged regions, we can evaluate the electrostatic compatibility across the interface. To evaluate electrostatic fields, we first identify a region within the potential field at some potential *p* called an electrostatic isopotential. The electrostatic isopotential is a subset of the potential field where the potential equals *p* at all points on the isopotential surface. By definition, this isopotential creates a threshold k (kT/e), where one side of the threshold has potential less than k and the other side has potential greater than k. Geometrically, an isopotential guarantees a geometric solid with measurable volume, which is the metric we use for evaluating influential amino acids.

#### 2.2.1. Interface Field Comparison

This implementation for calculating volumetric objects representing electrostatic fields has a unique utility of enabling volumetric comparisons. VASP-E manipulates isopotential surfaces with constructive solid geometry (CSG) techniques. CSG can calculate the union, intersection, and difference of volumetric objects [8]. Specifically, we generate fields representing the binding interface of proteins from both proteins in a protein complex. For one side of the interface of one protein, the isopotential surface is generated at +k, while on the other protein, a surface is generated at -k. The intersection between the +k and -k surfaces represents the region where the two protein complexes have opposite charges; accordingly, the intersection of +k and -k measures the degree of complementarity between opposite sides of the interface; a greater volume means higher electrostatic complementarity. The logic for intersection utility can be seen in Figure 2. Such a comparison measures even small-scale changes in electrostatic complementarity; for example, mutation of a single residue may exhibit an almost imperceptible change in charge compared to the full electrostatic field, however VASP-E is sensitive to these changes.

#### 2.2.2. Nullification

VASP-E is able to detect small-scale changes in electrostatic isopotential volume; exploiting this feature, we intentionally remove the charge contribution from single amino acids in a process called nullification [18]. Nullification of residues not only affects the electrostatic isopotential of the protein, but also measures the degree of the charge on the residue. A strongly positive residue that is nullified will exhibit a sharp decrease in charge after nullification. Although DelPhi disperses some of the change in charge across the whole protein, the change is usually negligible. Thus, we can assume that nullification of amino acids outside of the protein-protein interface region will result in negligible electrostatic change at the interface.

We compare the intersection of interface fields for individual amino acids in the protein complex to their nullified variant, mirroring lab experiments that compare wild type proteins to mutant proteins with a single point mutation. Figure 3 demonstrates the electrostatic volume change from nullifying a single residue in the Barnase-Barstar complex. Nullification of residue 59 removed a significant amount of electrostatic complementarity at the interface.

### 2.3. Validating DiffBond

#### 2.3.1. Dataset Construction

To demonstrate that DiffBond is able to identify and classify significant intermolecular ionic bonds, hydrogen bonds, and salt bridges, we validated DiffBond predictions against two datasets relating to two extensively studied protein complexes: Barnase-Barstar, Rap1a-raf, SMAD2-SMAD4, and complexes involving the three-finger toxin family. Data Set A contains extensively studied amino acid pairs that form intermolecular bonds for all four protein complexes. *Data Set B* is a summary of mutations that have been performed on Barnase-Barstar, Rap1a-raf and SMAD2-SMAD4, paired with the change in binding affinity due to mutation.

The structures in Data Set A consist of the Barnase-Barstar complex (pdb: 1brs), the Rap1A-raf complex (pdb: 1c1y), and the SMAD2-SMAD4 trimer (pdb: 1u7v). It also complexes between members of the three-finger toxin family with neuronal nicotinic acetylcholine receptors (nAChRs) (pdb: 1yi5, 4hqp, 2qc1, 1kc4). Each of these complexes have been extensively studied for specific electrostatically influential amino acids and intermolecular bond formations that affect their binding affinity.

Data Set B uses some of the structures in A (pdb: 1brs, 1c1y), but complements this structural data with 113 mutations of the proteins in these complexes (1brs(94), 1c1y(17)). These mutations, derived from the Structural database of Kinetics and Energetics of Mutant Protein Interactions (SKEMPI), are paired with experimentally measured changes in binding affinity that are caused by the mutation. Since no information on three-finger toxin proteins exists in SKEMPI and only two mutations were reported for SMAD2-SMAD4 complex, these complexes were omitted from Data Set B.

#### 2.3.2. Barnase-Barstar

Barnase is an extracellular ribonuclease of Bacillus amyloliquefaciens often co-expressed with its intracellular inhibitor barstar. Barnase can be lethal to the cell by itself, but is countered by forming a complex with barstar [19]. Barnase and Barstar form a tight complex with many intermolecular electrostatic interactions between residues at the binding site [20,21,22]. Experiments have demonstrated several influential residues involved in intermolecular interactions that result in enhanced or diminished electrostatic complementarity between Barnase and Barstar, as well as residues that exhibited no change in electrostatic complementarity.

#### 2.3.3. Rap1a-raf

Ras is a family of GTPase involved in transmitting signals to regular biological systems like cell cycle progression, cell division, apoptosis, lipid metabolism, DNA synthesis, and cytoskeletal organization [23]. Although the structures of ras in complex with many of its effector ligands are relatively unknown, Rap1a is a functional homolog to ras proteins and forms a complex with raf that is well studied. Rap1a binding site is almost identical to ras structures and binds competitively to raf, a ras effector oncogene involved in ERK 1/2 signaling [24,25]. The binding interface of Rap1a-raf consists of several significant intermolecular bond interactions whose mutations have been shown to alter binding affinity [24,26,27].

#### 2.3.4. SMAD2-SMAD4

SMADs is a family of proteins that act as main signal transducers for TGF-B receptors, a super family of proteins that help regulate cell development and growth [28,29]. Specifically, SMAD2 helps direct TGF-B signaling while SMAD4 mediates heteromeric complex formation between R-SMADs and SMAD4 [28]. The complex formation results in a trimer consisting of one SMAD4 with two SMAD2 subunits; the interface between each subunit in the trimer has been extensively surveyed for electrostatic interactions [30].

#### 2.3.5. Three-Finger Toxin Family

The three-finger toxins are a superfamily consisting of toxin proteins from elapid snake venom and similar structures [31,32]. Members of the family consist of three beta strand loops emanating from a cysteine rich core which form the distinct three-finger structure. Neurotoxin proteins like α-bungarotoxin [33,34,35] and α-cobratoxin [36] interact with nAChRs while other members can interact with different neuronal nAChR subtypes [32]. The fast and tight interaction between three-finger toxins and nAChRs consist of several well studied intermolecular interactions at the toxin-receptor interface. In addition, many of the three-finger toxins compete with each other for binding at known classical binding sites for agonists and competitive antagonists, making members of this superfamily an interesting set of proteins to study [37].

#### 2.3.6. Validation on DataSet A

On Dataset A, we validated DiffBond by comparing bond predictions to known protein-protein interactions published in experimental findings. Intermolecular bond formations were gathered from published journal papers where each bond was verified manually. We required two specific criteria to be met for confirming intermolecular bonds. First, the bond must verifiably exist between a pair of residues, and explicitly state the residue from each side of the interface. For salt bridges specifically, we considered existence of both a hydrogen bond and an ionic bond to imply existence of a salt bridge even if a salt bridge is not explicitly stated. Second, the paper must explicitly classify the bond as either an ionic bond, hydrogen bond, or salt bridge, or provide biochemical information that establishes one of the three bonds. Through this verification process, we can compile intermolecular bond information for protein complexes and compare this data to DiffBond prediction.

In our evaluation of DiffBond, we counted true positives (TPs), false positives (FPs), true negatives (TNs), and false negatives (FNs). A bond prediction is defined as a TP if DiffBond predicted the formation of a bond type between two specific amino acids, and experimental findings confirmed the same amino acid pair and bond type. A prediction is considered a FP if DiffBond predicted the formation of a bond, but experimental findings did not agree or did not mention the specific bond. TNs are bond predictions that were predicted to not occur by DiffBond and from experimental findings. FNs are evaluated as intermolecular bonds that DiffBond failed to predict when experimental findings confirmed the existence of the bond. From these measurements, we compute the precision and recall of DiffBond. Precision is the fraction of correctly predicted bonds among all bonds verified in experimental findings, and recall (sensitivity) is the fraction of correctly predicted bonds among all true interactions.

There are two practical issues when evaluating prediction performance using the above measurements. First, we cannot fully count TNs by nature of experiments in this field; no studies to our knowledge have exhaustively analyzed the electrostatic influence of every amino acid in a protein complex. Second, FPs as we measure them include bond predictions that may not have been experimentally tested before. As a result, FP matches may include intermolecular bonds that are yet to be found in future experiments. This conservative criteria for FP matches also implies that the reported precision of DiffBond is a lower limit of the true precision.

#### 2.3.7. Validation on DataSet B

A second way to validate the authenticity of intermolecular bonds predicted by DiffBond is to evaluate how their removal corresponds to binding. If the predicted bonds actually exist and play some role in protein-protein recognition, then we expect that, on balance, removing them through mutation should reduce affinity, and that affinity is maintained in mutants that do not remove the bonds. Thus, we used data corresponding to 113 mutants of the Barnase-Barstar and Rap1a-raf.

SKEMPI provides $K_{eq}$, the equilibrium rate of protein-protein interactions and a metric for binding affinity, but $K_{eq}$ provided are not easily comparable due to temperature differences in experiments. We normalized $K_{eq}$ to account for temperature differences and to view binding affinity with a logarithmic scale; this is done through a 2-step process. First, we compute the change in ∆G (∆∆G). This value is a measure of the change in energy for folded and unfolded states when a point mutation is present, and is a good predictor for whether a point mutation stabilizes the protein complex. Second, we normalize all ∆∆G grouped by protein complex to between 0 and 1; this avoids different scaling of free energy changes from complex to complex and allows us to compare ∆∆G across all protein complexes. In the context of protein interactions, a larger ∆∆G predicts that a protein interaction requires more energy input to stabilize the complex after the mutation, therefore suggesting an unfavorable mutation; a smaller ∆∆G means less energy input is necessary for binding to occur and a favorable mutation for protein-protein interaction.

DiffBond attempts to assess electrostatic complementarity through hypothetical changes in intermolecular bond formation due to mutations. An cartoon of the region that DiffBond analyzes is shown in Figure 2. As a result of DiffBond, predictions for intermolecular bond interactions at the interface are compiled. Interpreting each of these interactions according to their corresponding biochemical role provides a stronger prediction of binding affinity changes from mutational experiments. Salt bridges and ionic bonds, defined as an interaction between charged residues, are expected to affect electrostatic complementarity when mutated to an uncharged amino acid due to loss of bond formation. While hydrogen bonds are also a significant contributor to protein binding, hydrogen bonds occur frequently between charged and uncharged residues and so binding affinity changes are expected to be less correlated to electrostatic complementarity changes.

Volumetric comparison of interface fields and nullification has been shown to effectively predict binding affinity changes in mutational experiments [8,38]. Consequently, the predictions from this method are appropriate for validating DiffBond results and also for possibly refining predictions from DiffBond by utilizing both DiffBond and VASP-E methods in conjunction.

## 3. Results

### 3.1. Validation of DiffBond on Dataset A

The precision and recall of DiffBond predictions, on Barnase-Barstar, Rap1a-raf, SMAD2-SMAD4, and three-finger toxin-nAChR, are reported in Table 1. Ionic bonds, hydrogen bonds and salt bridges were separately counted. Although the total number of predictions for ionic bonds and salt bridges were low at n=16 and n=7 respectively, predictions exhibited high precision in general. Precision for predicting ionic bonds, hydrogen bonds, and salt bridges were 87.5%, 87.5%, and 85.7% respectively. Ionic bond and hydrogen bond prediction showed 82.4% and 74.5% recall, however salt bridge prediction had lower recall at 50%. The lowered recall is likely due to strict criteria used here and in previous studies [2,16] to precisely identify biochemically relevant salt bridges at the cost of missing verified salt bridges whose structure indicates possible biochemical unrelatedness. These findings indicate that most of the bond predictions made by DiffBond were verified in literature, giving high precision, and that most verified ionic and hydrogen bonds were predicted by DiffBond, giving high recall.

### 3.2. Validation of DiffBond on Dataset B

Figure 4 shows a comparison of bond predictions for ionic bonds, salt bridges, and hydrogen bonds to the Normalized ∆∆G computed for each bond prediction group. The first group contains the “Bond Broken” case, where DiffBond predicted that the bond exists before mutation but does not exist after mutation, inferring that a mutation would break a pre-existing bond. The second group contains the “Bond intact” case, where DiffBond predicted a bond exists before mutation with no change to bond type after mutation; this group specifically comprises mutations from one charged amino acid to another similarly charged amino acid. The third group contains the “Bond Formed” cases where DiffBond predicted a bond to not exist in the wildtype but exists after mutation.

It is important to note that amino acids in some bonds have alternate bonding partners, so the removal of one partner by mutation may result in bond formation with a third party. Although both ionic bonds and salt bridges are theoretically capable of forming in this way, Dataset B contained no mutations where this occurred. Figure 4 includes zero-value bars and an asterisk (*) for formation of ionic bonds and salt bridges to indicate this result.

We found that normalized average ∆∆G was significantly correlated to several groups of DiffBond predictions for whether a bond was broken or stayed intact due to mutation, as shown in Figure 4a,b. Ionic bonds and salt bridges showed large differences in normalized average ∆∆G for bond broken and bond intact groups for both Barnase-Barstar (1BRS) and Rap1a-raf (1C1Y). A significantly higher normalized average ∆∆G for bond broken group implies a mutation that breaks an ionic bond or salt bridge is unfavorable for affinity. The confidence intervals for salt bridge predictions overlap slightly with normalized average ∆∆G having an overlap of (0.516, 0.664) and (0.283, 0.530) for “Bond Broken” and “Bond Intact” respectively. However, the slight overlap may also be due to high variance of normalized average ∆∆G when bonds are predicted to be intact. Variance is especially high for Rap1a-raf (pdb: 1C1Y), as seen in Figure 4 and Figure 5, likely due to smaller sample sizes of n=16, n=9, and n=9 for ionic bond, hydrogen bond, and salt bridge predictions respectively. Rap1a-raf had only one prediction made where hydrogen bonds are broken and none for hydrogen bonds forming due to mutation. As a result, no variance can be calculated, but normalized average ∆∆G is provided for predictions where hydrogen bonds stay intact. In general, DiffBond had low variance in normalized average ∆∆G when predicting ionic and salt bridges breaking (avg CV=0.323) compared to ionic and salt bridge bonds intact (avg CV=0.705).

DiffBond predictions on hydrogen bond breaking, staying intact, or forming are much less correlated to normalized average ∆∆G than those of ionic bonds and salt bridges (Figure 4c). First, confidence intervals across almost all groups overlapped, except Rap1a-raf (1C1Y) “Bond Broken” which contained one data point and “Bond Formed” which contained no points, which is insufficient data to calculate an interval. We found that normalized average ∆∆G was also lower for when breaking a hydrogen bond compared to the other groups, which had similar normalized average ∆∆G to each other. Hydrogen bonds are involved in salt bridges; we did not separately evaluate hydrogen bonds not involved in salt bridges so that methods for identifying hydrogen bonds and salt bridges are independent. Statistics of normalized average ∆∆G for all groups are included in Appendix A.

In addition to measuring normalized average ∆∆G, we also use volumetric change in interface fields from VASP-E as a metric for change in electrostatic complementarity as shown in Figure 5. By comparing to VASP-E, we demonstrate DiffBond capability to predict electrostatic complementarity rather than protein complex stability based on ∆∆G. For a protein complex, we compute the intersection of the oppositely charged fields at the interface such that we get two intersection volumes at *k* = +1/−1 and *k* = +5/−5 for the wildtype complex and mutant complex. This results in four intersection volumes of the interface: +1/−1 for wildtype and mutant, and +5/−5 for wildtype and mutant. Each intersection volume indicates the amount of electrostatic complementarity between the two proteins in the protein complex. As a result, a large value means greater complementarity and vice versa. By computing the difference in volumes, $v(WT_{k}-v\left(Mut_{k}\right)$, we can assess the change in complementarity due to mutation; a larger wildtype intersection volume compared to mutant would result in a positive volume, indicating lowered complementarity due to mutation, while a negative volume indicates increased complementarity. Finally, we compute the sum of the differences for *k* = +1 and −1 at the interface and normalize by group to summarize total complementarity change for positive and negative charge. By interpreting volume changes as changes in electrostatic field at the protein-protein interface, the resulting sum of interface intersection illustrates changes in electrostatic complementarity due to mutation.

We compared electrostatic intersection volume differences computed by VASP-E for the two groups, “Bond Broken” and “Bond Intact”, to assess DiffBond predictions (Figure 5). We found that the intersection difference was larger for when a bond is broken than when a bond is intact, especially for Barnase-Barstar. A large change in intersection difference suggests that electrostatic complementarity is lower since wildtype electrostatic complementarity is much larger than mutation electrostatic complementarity. Rap1a-raf (1C1Y) showed slight overlap between intersection difference at *k* = +5/−5 of “Bond Broken” and “Bond Intact” for both ionic bonds and salt bridges. However, for predicting volume difference of ionic bond for the same Rap1a-raf groups there is a significant difference, contrasting with ∆∆G results. For all other groups, DiffBond prediction shows significant differences in volume change for broken bonds compared to intact bonds. Although hydrogen bonds are important bonds in protein stability, nullification does not remove pre-existing hydrogen atoms from the side chain, only ignores the charges that exist. Since hydrogen bonds are unchanged before and after nullification, we exclude hydrogen bond comparisons when examining electrostatic complementarity with VASP-E. Statistics of interface intersection volume changes for all groups are included in Appendix A.

## 4. Discussion

We present DiffBond, a novel method for identifying and classifying intermolecular bonds in protein-protein interactions. DiffBond was designed to gather structural and electrostatic information about bond formation in a biological environment through exhaustive computational analysis of a protein complex to predict influential amino acids for binding. The prediction of influential amino acids towards guiding mutational experiments is relevant in fields like protein engineering and drug design. By identifying key residues for binding, we can form hypotheses about the mechanism by which proteins interact, whether through key electrostatic bonds like ionic bonds, salt bridges, and hydrogen bonds or through other processes involving steric hindrance, electrostatic fields, hydrophobic interactions. This knowledge is key to deciding mutations that modify protein binding specificity towards a desired effect, whether towards increasing the medicinal effects or reducing harmful effects of target proteins.

Identification and classification of salt bridges, ionic bonds, and hydrogen bonds using DiffBond is a novel approach that demonstrated promising capabilities through validation on a small curated data set. DiffBond predicted ionic bond and hydrogen bond formation with 87.5% precision for both ionic bond and hydrogen bond predictions and 82.4% and 74.5% recall for ionic bond and hydrogen bond predictions respectively. Similarly, salt bridge prediction maintained 85.7% precision, and a lower recall than ionic and hydrogen bonds at 50%. Although 50% recall of intermolecular salt bridges indicates that half of the verified salt bridges in literature were not predicted by DiffBond, precision for all three bonds, which was a conservative estimate of true precision, shows that DiffBond rarely predicts a bond to exist when it does not. Overall, DiffBond was effective in correctly predicting the presence of 55 bonds and only incorrectly predicting 8. As a bond prediction tool that can be incorporated into an Analytic Ensemble, strong performance in precision is preferable over performance in recall, so that incorrect predictions infrequently waste analytical effort. Incorporation of DiffBond into an Analytic Ensemble will allow other related methods to add interaction information towards improving recall.

On validation with experimentally determined binding affinities from SKEMPI, DiffBond predicted whether a bond breaks or stays intact due to mutation. We found that this prediction was significantly associated with changes in binding affinity using ∆∆G (Figure 4) and with changes in electrostatic complementarity (Figure 5) at the interface of protein-protein interaction. This is especially true for ionic bonds, which clearly differentiated protein stability and electrostatic complementarity based on predictions for almost all groups examined. Salt bridges had lower significance when differentiating ∆∆G values by prediction, but still had weak correlations for higher ∆∆G and lower electrostatic complementarity when bonds are broken, indicating decreased binding affinity and decreased stability of the protein complex. Hydrogen bonds are very weakly correlated to ∆∆G, but our concern with hydrogen bonds is more so its influence as part of a salt bridge in affecting protein-protein interactions rather than as an independent bond, and we found that salt bridges correspond much more with ∆∆G and electrostatic complementarity changes. Overall, DiffBond predicted higher ∆∆G for a broken bond compared to a bond that remained intact for Barnase-Barstar and Rap1a-raf across all ionic bond and salt bridge mutations, demonstrating the ability of DiffBond to associate intermolecular bonds to increased binding specificity.

DiffBond, as a bond prediction method, is most applicable as a preliminary search for significant residues in a protein structural model or as part of an Analytic Ensemble approach. Mutational experiments rely on precise algorithms for identifying significant residues to guide mutation testing; experiments are often limited in number of mutations that can be tested and so precise prediction methods can help reduce unproductive testing. In addition, structural biologists can benefit greatly from transparent implementation and explainable results. DiffBond was implemented based on the chemical definition of bonds formed between the 20 canonical amino acids at biological pH, and predictions are defined simply as whether the software identified intermolecular bonds in the crystal structure based on the bond definition. As a result, DiffBond produces explainable results which allow us to make specific inferences about the biochemical mechanism for protein binding and specificity. This design also synergizes with other approaches to analyzing protein binding. DiffBond used in conjunction with other methods in an Analytic Ensemble that identify influential electrostatic interactions may provide a more comprehensive summary of the common electrostatic interactions across the binding interface in protein-protein interactions. Further extension of DiffBond in the direction of Analytic Ensemble work and generalizability to molecules other than proteins at biological pH is a topic of interest.

The capability to computationally generate biochemical explanations for the role that a specific mechanism plays in binding is novel to the analytic ensemble strategy. In contrast to the analytic ensemble approach, conventional methods like molecular docking predict structure of proteins and complexes, but do not make inferences on mechanisms involved in binding. In such cases, inferences about influential mechanisms require human experts to interpret results. These capabilities point to new applications in the analysis of protein structure and in interpreting the biochemical mechanisms in predicted structures, thereby enhancing the design of mutational experiments aimed to elucidate protein structure and the comprehensibility of computational software outputs.

## Figures and Tables

**Figure 1 molecules-27-06178-f001:**
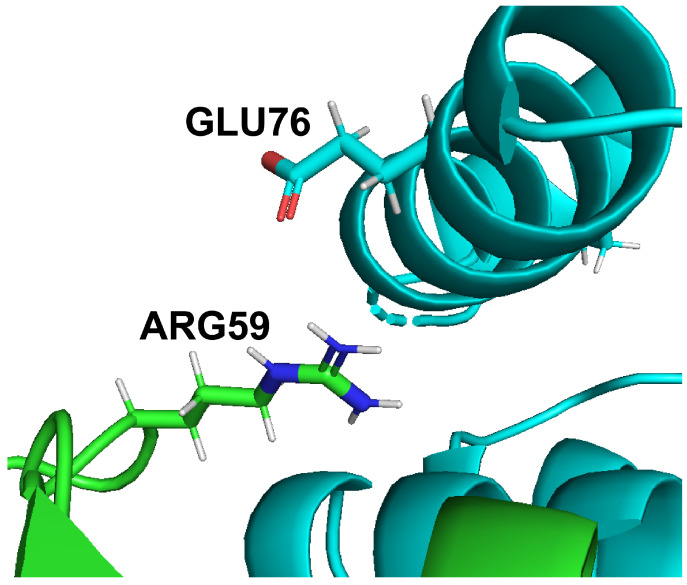
Sidechain visualization of Arg59 on Barnase (green) and Glu76 on Barstar (teal). Arg59 and Glu76 are within 4 Å and are oppositely charged amino acids, so they are predicted to form a salt bridge by DiffBond.

**Figure 2 molecules-27-06178-f002:**
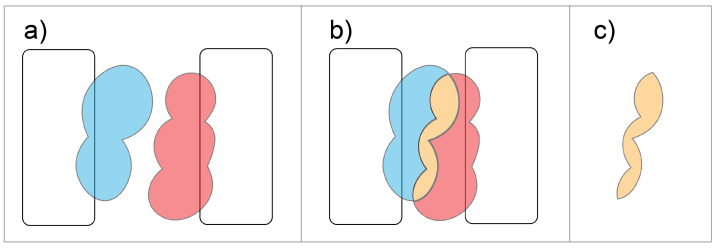
Intersection using CSG involves: (**a**) Two proteins with oppositely charged electrostatic fields. (**b**) When the proteins are in complex, the oppositely charged fields overlap forming an intersection region shown in orange. (**c**) The intersection region represents the degree to which the field of one protein complements the field of the other.

**Figure 3 molecules-27-06178-f003:**
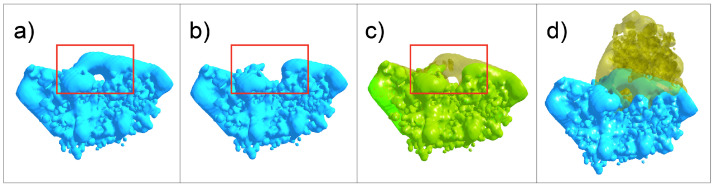
Effect of Nullification on Barnase-Barstar. (**a**) Wildtype Barnase electrostatic surface at isopotential of +1 kT/e. (**b**) Barnase nullified at residue 59, electrostatic surface at isopotential of +1 kT/e. (**c**) Overlap of wildtype (transparent yellow) and nullified Barnase (green) surfaces. (**a**–**c**) The red square encompasses the main difference in isopotential surface due to nullification. (**d**) Wildtype Barnase (blue) in complex with Barstar (transparent yellow).

**Figure 4 molecules-27-06178-f004:**
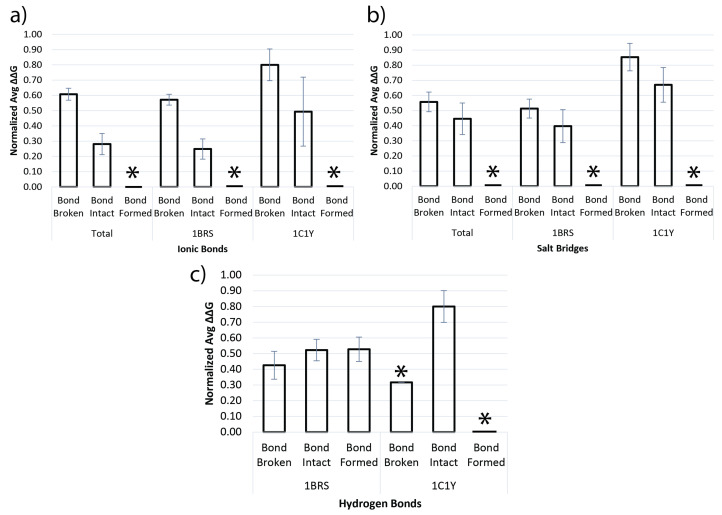
Comparison of ∆∆G when a bond is broken and when a bond remains intact for (**a**) Ionic bonds, (**b**) Salt Bridges, and (**c**) Hydrogen bonds. Each bond compared using normalized average ∆∆G values for Barnase-Barstar (1BRS) and Rap1a-raf (1C1Y). (*) indicates that no bonds were predicted by DiffBond in those groups. Rap1a-raf hydrogen bond had only one bond broken prediction, so no intervals were calculated.

**Figure 5 molecules-27-06178-f005:**
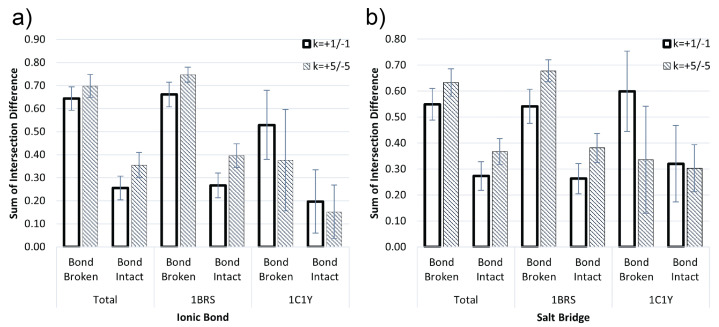
Comparison of interface intersection volume changes when a bond is broken and when a bond remains intact for (**a**) Ionic bonds and (**b**) Salt Bridges. Each bond compared using volumes calculated from VASP-E for Barnase-Barstar (1BRS) and Rap1a-raf (1C1Y).

**Table 1 molecules-27-06178-t001:** Precision and Recall of the bond list for predicting the formation of bonds.

	Ionic Bond	Hydrogen Bond	Salt Bridge
True Positive	14	35	6
False Positive	2	5	1
False Negative	3	12	6
True Negative	Unknown	Unknown	Unknown
Precision	87.5%	87.5%	85.7%
Recall	82.4%	74.5%	50.0%
Total Known Bonds	17	47	12
Total Predictions	16	40	7

## Data Availability

Publicly available datasets were analyzed in this study. This data can be found here: https://life.bsc.es/pid/skempi2 (accessed on 10 September 2022). DiffBond software, datasets used in this work, and supplemental data can be found here: https://github.com/LehighInfolab/DiffBond (accessed on 10 September 2022).

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
