# Peer review of "Analysis of Protein-Protein Interactions for Intermolecular Bond Prediction"

_molecules, 2022, doi:10.3390/molecules27196178_

Round 1

Reviewer 1 Report

The manuscript by Tam et al. describes the DiffBond tool for predicting intermolecular bonds (ionic bonding, salt bridge, and hydrogen bonding) in four protein complexes, the manuscript is well written, easy to read and understand, and the results are relevant and demonstrate the benefits of DiffBond as a prediction tool.

The authors should discuss in depth the results of table 1

It is recommended to include section “5. Conclusions” and include the most relevant contributions of the research

Figure 5. Why 1C1Y has wide confidence intervals? The authors should discuss this result in depth

Reviewer 2 Report

In this paper the authors expand a previous work on a method for assessing the predictability of intermolecular bonds. The authors claim that their program DiffBond is the first to predict intermolecular bond interactions at the binding interface between protein-protein interactions where it classifies all interactions as salt bridges, ionic bonds, or hydrogen bonds. From reading the paper it did not become apparent to me what the practical purpose of this classification could be. Moreover, salt bridges should be easily seen in crystal structure or are evident from the surface charge of a given protein. Evidently, it does not allow us to calculate free energies of binding. Hence, it is unclear for whom this program may be useful. In its present state the paper is thus a technical report at best. I suggest that the authors should at least bother to explain non-experts what this program can do beyond well-established packages as e.g. DelPhi.
